# Effect of Dexmedetomidine on Cardiopulmonary Bypass Induced Inflammatory Response in Patients Undergoing Aortic Valve Replacement

**DOI:** 10.3390/life15040524

**Published:** 2025-03-22

**Authors:** Zrinka Safaric Oremus, Nikola Bradic, Ivan Gospic, Ivana Presecki, Sanja Sakan, Natasa Sojcic, Kresimir Oremus, Davor Baric, Vlatka Sotosek, Igor Rudez

**Affiliations:** 1Department of Anaesthesiology, Reanimatology and Intensive Care Medicine, Dubrava University Hospital, 10000 Zagreb, Croatia; nbradic@kbd.hr (N.B.); ivan.gospic82@gmail.com (I.G.); ivanapresecki100@gmail.com (I.P.); sakan1981@gmail.com (S.S.); natasa.sojcic@gmail.com (N.S.); 2Department of Anaesthesiology, Akromion Special Hospital for Orthopedic Surgery, 10000 Zagreb, Croatia; kresimir.oremus@akromion.hr; 3Department of Cardiac and Transplant Surgery, Dubrava University Hospital, 10000 Zagreb, Croatia; dbaric@kbd.hr (D.B.); rudi@kbd.hr (I.R.); 4Department of Anesthesiology, Reanimatology, Emergency and Intensive Care Medicine, Faculty of Medicine, University of Rijeka, 51000 Rijeka, Croatia; vlatkast@medri.uniri.hr

**Keywords:** aortic valve stenosis, aortic valve replacement, cardiopulmonary bypass, cytokines, dexmedetomidine, inflammation, interleukine-6, tumor necrosis factor-alpha

## Abstract

Surgical aortic valve replacement (SAVR) remains an essential treatment option for patients with aortic stenosis (AS). Open-heart surgery requires the use of cardiopulmonary bypass (CPB), which triggers an inflammatory response that can lead to end-organ dysfunction and severe complications. Dexmedetomidine, a highly selective α2-adrenergic agonist, is widely used in anesthesia and intensive care medicine for its sedative, analgesic, and sympatholytic properties. This study aimed to investigate whether dexmedetomidine exerts a clinically relevant anti-inflammatory effect in patients undergoing open-heart surgery and to determine the optimal dose. A prospective, double-blind, placebo-controlled study was conducted, including 60 patients randomized into three groups according to dexmedetomidine dose. Inflammatory markers (IL-6, TNF-α), renal function, and other clinical parameters were analyzed at multiple time points. Statistical analyses were performed to assess differences between the groups. Dexmedetomidine administration significantly affected TNF-α levels 12 h after CPB (*p* = 0.033), while previously reported suppression of IL-6 was not observed. Dexmedetomidine was associated with lower opioid consumption before extubation and showed a tendency to reduce postoperative delirium. Diuresis was significantly increased on the first postoperative day in dexmedetomidine-treated patients (*p* = 0.003), with no significant changes in other renal parameters. The incidence of atrial fibrillation was highest in the control group and lowest in the high-dose dexmedetomidine group, though this difference was not statistically significant. These results suggest that dexmedetomidine influences inflammatory and clinical outcomes; however, further research is needed to confirm its long-term benefits and optimal dosing strategies.

## 1. Introduction

Aortic stenosis (AS) is one of the most common and severe valvular heart diseases affecting older adults [1,2]. It is characterized by the narrowing of the aortic valve, which impedes blood flow from the left ventricle to the aorta, leading to increased cardiac workload, left ventricular hypertrophy, and, eventually, heart failure, if left untreated. Majority of morbidity and mortality attributable to valvular heart disease worldwide, including AS, is due to rheumatic heart disease, which is commonly seen as a cause of AS in low-income countries. In high-income countries, calcific aortic valve disease (CAVD) is the prevalent cause of AS [2,3]. The disease primarily stems from progressive calcification of the valve, often exacerbated by age-related factors, metabolic conditions, and congenital anomalies such as bicuspid aortic valves [2,3]. Globally, the prevalence of AS increases significantly with age. In individuals aged 75 years and older, the prevalence ranges between 9% and 13%, making it a leading cause of morbidity in elderly populations [2,4]. The Global Burden of Disease study highlights that in 2021, calcific aortic valve disease (CAVD) contributed to 2.24 million disability-adjusted life years (DALYs), marking a 30.5% increase since 1990 due to aging populations [3]. Rising prevalence of valvular heart disease associated with advancing age combined with rapid aging of populations worldwide has labeled valvular heart diseases as the “next cardiac epidemic” [5,6]. Additionally, individuals with congenital anomalies, such as bicuspid aortic valves, are predisposed to earlier onset of AS [2,7]. The number of aortic valve procedures performed over the past decades has been constantly increasing, while at the same time, the number of mitral valve surgeries remains constant.

Cardiopulmonary bypass (CPB) is an essential technique in cardiac surgery that temporarily takes over the functions of the heart and lungs, enabling surgeons to perform complex procedures under controlled conditions. However, the benefits of CPB come at the cost of a significant systemic inflammatory response syndrome (SIRS), which can lead to multi-organ dysfunction, postoperative complications, and prolonged recovery. This inflammatory response arises from blood contact with non-endothelial surfaces, ischemia–reperfusion injury, and activation of various immune and coagulation pathways [8,9,10,11,12]. Monocytes and endothelial cells produce pro-inflammatory cytokines, including interleukin-6 (IL-6), tumor necrosis factor-alpha (TNF-α), IL-8, and IL-1β, which amplify the inflammatory cascade [8,9,10]. Neutrophils adhere to the endothelium, release reactive oxygen species (ROS) and proteolytic enzymes, and contribute to microvascular injury and capillary leak syndrome. The second phase involves ischemia–reperfusion injury, which occurs when blood flow is restored after a period of ischemia, particularly to the myocardium, lungs, kidneys, and intestines [8,12]. This phase triggers oxidative stress, increased cytokine production, and further endothelial injury, perpetuating the inflammatory response. Reactive oxygen species exacerbate endothelial dysfunction, increase vascular permeability, and damage cellular structures [8,9,12]. CPB also results in the shedding of the glycocalyx layer, impairing vascular integrity and promoting edema and inflammation. IL-6 is a central pro-inflammatory cytokine and a key marker of the inflammatory response during CPB. Elevated IL-6 levels correlate with postoperative complications, including impaired lung function, circulatory instability, and organ dysfunction [8,9]. TNF-α is an early pro-inflammatory cytokine that promotes leukocyte recruitment, vascular permeability, and tissue injury. Although TNF-α levels during CPB are generally lower than IL-6, elevations are associated with poor outcomes [8,9,10,12]. The cytokine surge and endothelial activation contribute to the development of SIRS, characterized by vasoplegia, often requiring vasopressors, pulmonary dysfunction due to increased alveolar-capillary permeability resulting in acute lung injury or ARDS, acute kidney injury (AKI) caused by ischemia, oxidative stress, and inflammation, and coagulopathy and bleeding due to platelet dysfunction, fibrinolysis, and disruption of the coagulation cascade [8,9,12,13]. It is estimated that approximately 1.5 to 2 million CPB procedures are performed each year as part of open-heart surgeries [14]. Despite advances in biocompatible CPB circuits and minimally invasive techniques, inflammation remains a challenge.

Dexmedetomidine is an α2-adrenergic agonist commonly used in anesthesia and intensive care settings. It is potent and highly selective for α2 adrenergic receptors with α2: α1 ratio of 1620:1 [15]. Dexmedetomidine acts on α2-adrenergic receptors in the central nervous system, particularly in the locus coeruleus, resulting in sedation that resembles natural sleep. Its sympatholytic effects reduce heart rate and blood pressure, while its analgesic action is mediated through both spinal and supraspinal mechanisms. Dexmedetomidine also influences the cholinergic anti-inflammatory pathway, which may play a role in mitigating the systemic inflammatory response to surgical trauma and ischemia–reperfusion injury [16]. Even though it has been approved for sedation of adult intensive care unit patients [17], dexmedetomidine is widely used as an adjunct to general anesthesia and as a sedative agent in patients undergoing procedures under regional anesthesia. Its opioid-sparing effect is particularly valuable in multimodal analgesia protocols aimed at reducing opioid consumption and minimizing opioid-related side effects, such as respiratory depression, nausea, and postoperative delirium [18]. Dexmedetomidine has demonstrated cardioprotective properties in ischemia–reperfusion models, reducing myocardial injury and improving cardiac function [19]. Preclinical studies suggest that dexmedetomidine protects renal function by activating the cholinergic anti-inflammatory pathway and reducing oxidative stress [16,20,21,22]. Dexmedetomidine’s neuroprotective effects are particularly useful, especially in elderly patients undergoing cardiac and non-cardiac surgery. It attenuates neuroinflammation, reduces apoptosis, and promotes anti-oxidative responses, thereby protecting against ischemic brain injury [23].

Although substantial evidence supports the beneficial effects of dexmedetomidine in the cardiac surgery setting [12,19,23,24,25,26,27], definitive recommendations regarding the optimal timing, dosage, or specific patient cohort that would benefit most from its effects are still lacking. Our study aimed to evaluate whether dexmedetomidine exerts a clinically relevant anti-inflammatory effect in patients undergoing open-heart surgery and to determine the optimal dose.

## 2. Materials and Methods

### 2.1. Patient Selection and Study Design

We conducted a double-blind, randomized, prospective study enrolling 60 adult patients with isolated aortic stenosis (AS) scheduled for surgical aortic valve replacement (SAVR) at our institution between November 2022 and November 2023 (ClinicalTrials.gov (accessed on 1 February 2025) identifier: NCT05641064). All patients provided written informed consent. The study was approved by the Ethics Committee of University Hospital Dubrava, Zagreb, Croatia (ID 2021/0211-01), in accordance with the Ethical Principles for Medical Research Involving Human Subjects outlined in the Declaration of Helsinki by the World Medical Association. Patients were randomized into three groups using a sealed envelope method. Randomization was conducted by an anesthesia technician who was not involved in perioperative patient care or data collection. The same technician supervised the preparation of the infusion solutions. The infusion pump settings were concealed from the operating room (OR) staff. Group 0 served as the control group and received a saline infusion, Group 1 received dexmedetomidine as a continuous infusion at a dose of 0.5 μg/kg/h, and Group 2 received dexmedetomidine at a dose of 1 μg/kg/h. Tested solution infusion was initiated at the beginning of the surgical procedure. The randomization number was stored in a sealed opaque envelope until the conclusion of the study.

Patients were excluded if they met any of the following criteria: body mass index (BMI) > 30 kg/m^2^, first-, second-, or third-degree atrioventricular (AV) block, bradycardia < 50/min upon admission to the operating room (OR), neurological disorders (Parkinson’s disease, myasthenia gravis, multiple sclerosis, history of brain tumors), substance or alcohol abuse, current psychotropic drug therapy, or type I diabetes with complications. Patients who developed severe hypotension requiring vasoconstrictors following the initiation of dexmedetomidine infusion were also excluded. Any change in the surgical plan, discontinuation of the procedure, or violation of the intraoperative protocol constituted additional exclusion criteria. Patients who received corticosteroids during the perioperative period were also excluded. Baseline hematological, biochemical, hormonal (cortisol, ACTH (adrenocorticotropic hormone) and cytokine (IL-6 and TNF-α) assays were performed upon hospital admission.

### 2.2. Study Protocol and Data Collection

#### 2.2.1. Anesthesia and Surgical Procedure

All patients were premedicated with 0.1 mg/kg of morphine (Morfinklorid Alkaloid, ALKALOID-INT d.o.o., Ljubljana, Slovenia) intramuscularly 30 min prior to OR admittance. After patient identification upon arrival to the OR, standard non-invasive monitoring (ECG and bispectral index—BIS—Draeger Infinity BISx SmartPod, Luebeck, Germay) was applied and two large bore venous access lines were placed. Attending anaesthesiologist placed an arterial line after subcutaneous lidocaine injection for invasive blood pressure monitoring and blood sampling. After period of preoxygenation with 100% O_2_, anesthesia was induced with propofol (Propofol-Lipuro, B. Braun Melsungen AG, Melsungen, Germany) 1.5–2 mg/kg and sufentanil 0.4 μg/kg (Sufentanil Altamedics, Laboratoire Renaudin, Itxassou, France) followed by a bolus of 0.6 mg/kg rocuronium (Rokuronijev bromid, B. Braun Melsungen AG, Melsungen, Germany) to facilitate tracheal intubation followed by 10 mcg/kg/min continuous infusion until the end of surgery. After intubation, central venous catheter and pulmonary artery catheter were placed. Intraoperative monitoring included invasive blood pressure monitoring (iBP), 5 channel ECG, BIS, capnography, pulse oximetry (SpO_2_) (all monitoring: Draeger Medical Systems, Inc., Danvers, MA, USA), transoesophageal echocardiography (TEE) (Vivid E95, GE Vingmed Ultrasound, Horten, Norway). Anesthesia was maintained with sevoflurane (Sevofluran Baxter, Baxter S.A., Lessines, Belgium) in mixture of 50% oxygen in air to keep BIS value between 40 and 60 with additional sufentanil boluses according to clinical observations. At the end of the procedure the amount of used was noted.

SAVR was performed through a full median sternotomy under normothermic arrest, using crystalloid cardioplegia. Following systemic heparinization, the ascending aorta and right atrium were cannulated, and CPB was initiated. A left ventricular vent was placed through the upper right pulmonary vein, advancing trans-mitral into the left ventricular apex. Diastolic arrest was induced following aortic cross-clamping, with antegrade cardioplegia delivered via a cannula placed in the aortic root or directly into the coronary ostia. Valve replacement was performed through an oblique aortotomy. After leaflet excision and meticulous annular decalcification, mattress sutures with pledgets were placed on the ventricular side. Following annular sizing, a prosthetic valve was implanted in supra-annular position. The aortotomy was closed in two layers, with simultaneous deairing of the left ventricle and ascending aorta. After aortic clamp removal and reperfusion, the patient was weaned from CPB. Residual heparin was neutralized with protamine. Heparin and protamine dosing was guided intraoperatively using the Hepcon HMS Plus system (Medtronic, Minneapolis, MN, USA) to minimize the risk of heparin or protamine overdosing, thereby reducing the likelihood of postoperative bleeding and transfusion requirements. Valve function was assessed intraoperatively using TEE.

#### 2.2.2. Postoperative Treatment and Blood Sampling

Following surgery, patients were transferred to the intensive care unit (ICU) for further management. The incidence of bradycardia requiring temporary cardiac pacing (TCP) upon exit for the OR, as well as postoperative atrial fibrillation, was recorded. Standard hematological, biochemical, and coagulation assays were performed upon ICU admission and subsequently according to standard ICU protocol. A 12-lead ECG was obtained upon ICU admission. Chemiluminescence immunoassay was used to measure cortisol (DxI 800, Beckman Coulter, Brea, CA, USA) and adrenocorticotropic hormone (ACTH) (Maglumi 800, Snibe, Shenzhen, China) levels 24 h postoperatively to assess the effect of dexmedetomidine on the surgical stress response [22]. C-reactive protein (CRP) and procalcitonin (PCT) were measured daily. Blood samples for IL-6 and TNF-α were collected at four time points: upon hospital admission (T0), before CPB initiation (T1), 5 h after CPB termination (T2), and 12 h after CPB termination (T3). Blood samples were centrifuged at 600× *g* for 5 min, and the resulting serum was transferred into Eppendorf tubes and stored at −80 °C until the end of the study. Cytokine concentrations were measured using enzyme-linked immunosorbent assay (ELISA) kits (Invitrogen Human IL-6 ELISA Kit and Invitrogen Human TNF-α ELISA Kit, Thermo Fisher Scientific, Waltham, MA, USA; Bender MedSystems GmbH, Vienna, Austria).

Postoperatively, patients received morphine analgesia until extubation, and the total dose was recorded. After extubation, nonsteroidal analgesics were administered according to the protocol, including metamizole (Alkagin, ALKALOID-INT d.o.o., Ljubljana, Slovenia) (up to 5 g IV/day), paracetamol (Paracetamol Kabi, Fresenius Kabi Deutchland GmbH, Bad Homburg, Germany) (up to 3 g IV/day), and pethidine (Dolsin, BB Pharma a.s., Prague, Czech Republic) intramuscularly for severe pain. Extubation time was recorded. Postoperative delirium was assessed daily using the Confusion Assessment Method (CAM) questionnaire [28] in both the ICU and the surgical ward. The incidence of postoperative atrial fibrillation (AF), infection, and antibiotic use was recorded. Patients were observed throughout their hospital stay.

### 2.3. Statistical Analysis

All statistical analyses were conducted using SPSS software (version 20, IBM Corp., Armonk, NY, USA). Data distribution was assessed for normality using the Shapiro–Wilk test. Continuous variables were reported as mean ± standard deviation (SD) or median with interquartile range (IQR), depending on the data distribution. Categorical variables were expressed as absolute numbers and percentages. Between-group comparisons for continuous variables were performed using the one-way analysis of variance (ANOVA) for normally distributed data or the Kruskal–Wallis test for non-normally distributed data. A repeated-measures MANOVA was conducted to assess the differences between groups across multiple time points. For comparisons between two independent non-normally distributed groups, the Mann–Whitney U test was used. Categorical variables were analyzed using the chi-square test (χ^2^) or Fisher’s exact test, as appropriate. Atrial fibrillation incidence, infection rates, and postoperative delirium (CAM scores) were compared using the chi-square test with post hoc Bonferroni correction for multiple comparisons. Linear-by-Linear Association test was used to analyze trends in ordinal variables, such as increasing dexmedetomidine dose and its potential effects on CAM incidence, infection rates, and inflammatory marker levels over time. Least Significant Difference (LSD) test was used for post hoc comparisons in cases where ANOVA showed a significant difference among groups. A *p*-value < 0.05 was considered statistically significant for all analyses.

## 3. Results

A total of 60 patients were included in the study and randomized using the sealed envelope method into three groups of 20. Group 0 served as the control and received a saline infusion, Group 1 received dexmedetomidine at 0.5 μg/kg/h as a continuous infusion during the surgical procedure, and Group 2 received dexmedetomidine at 1 μg/kg/h as a continuous infusion. The baseline characteristics, including age and body mass index (BMI), were similar across the three groups, with no statistically significant differences (Table 1). Due to the nature of the randomization technique, sex distribution was slightly imbalanced.

### 3.1. Intergroup Differences in CPB Duration and Aortic Cross Clamp Time, Need for TCP, Opioid Consumption and Extubation Times

The differences in sufentanil consumption, CPB duration, aortic cross-clamp time, and extubation time among the groups are presented in Table 2. The total intraoperative dose of sufentanil was comparable across groups, with no statistically significant differences observed (*p* = 0.744). The mean doses were 93 ± 31.93 μg in Group 0, 87.5 ± 15.17 μg in Group 1, and 91.25 ± 18.63 μg in Group 2. Although a slight trend toward lower sufentanil consumption was noted in Group 1, this difference was not statistically significant.

The duration of cardiopulmonary bypass (CPB) was significantly longer in Group 2 (68.65 ± 14.74 min) compared to Group 1 (58.5 ± 11.78 min, *p* < 0.05). CPB duration in Group 0 was 64.55 ± 10.86 min, showing no significant difference compared to the other groups.

Bradycardia requiring temporary cardiac pacing (TCP) was rare in all groups, with only one patient in Group 2 requiring pacing. Given this low incidence, no statistically significant differences were found between the groups.

Postoperative morphine consumption until extubation was significantly lower in Group 2 (10.42 ± 6.82 mg) compared to Group 0 (17.25 ± 3.8 mg, *p* < 0.05) and Group 1 (14.25 ± 6.13 mg, *p* < 0.05).

Extubation time was similar across all groups, with no statistically significant differences observed (*p* = 0.261). The mean times were 8.38 ± 3.58 h in Group 0, 8.35 ± 2.81 h in Group 1, and 9.94 ± 3.91 h in Group 2. Although Group 2 had the numerically longest extubation time, this difference was not statistically significant.

### 3.2. The Effect of Dexmedetomidine on IL-6 and TNF-α Concentrations

IL-6 concentration increased significantly over time in all groups following CPB, confirming the activation of a systemic inflammatory response (*p* < 0.001 for time effect) (Figure 1a). At 5 h post-CPB (T2), the highest mean IL-6 concentration was observed in Group 2 (43.2 ± 40.27 pg/mL), followed by Group 1 and Group 0; however, the differences between groups were not statistically significant (*p* = 0.187). Up to 12 h post-CPB (T3), IL-6 levels remained elevated across all groups, with no significant differences between groups (*p* = 0.962) (Figure 1a).

Five hours post-CPB (T2), TNF-α concentrations tended to be lower in Group 1 (4.38 ± 3.33 pg/mL) than in Group 0 (6.17 ± 4.3 pg/mL) and Group 2 (7.06 ± 5.39 pg/mL), but this difference was not statistically significant (*p* = 0.167). At 12 h post-CPB (T3), TNF-α levels increased in Group 2 (6.81 ± 4.09 pg/mL), which was significantly higher compared to Group 0 (4.34 ± 3.09 pg/mL) and Group 1 (4.13 ± 3.00 pg/mL) (*p* = 0.033). Time had an overall significant effect on TNF-α levels (*p* = 0.005), although no statistically significant differences were observed between groups at most time points, except at T3 (Figure 1a).

### 3.3. The Effect of Dexmedetomidine on Cortisol and ACTH Levels

Cortisol levels increased significantly postoperatively in all groups, reflecting the expected activation of the hypothalamic–pituitary–adrenal (HPA) axis due to surgical stress. However, Group 2 had lower postoperative cortisol levels (653.25 ± 329.73 nmol/L) than group 0 (815.95 ± 326.59 nmol/L) and Group 1 (817.2 ± 274.76 nmol/L), although this difference was not statistically significant (*p* = 0.169) (Figure 2).

ACTH levels decreased postoperatively in all groups, probably due to negative feedback from increased cortisol levels. However, no significant differences were found among the groups (*p* = 0.570) (Figure 3).

### 3.4. Renal Function Parameters

Postoperative diuresis, renal function indices, and fluid balance were evaluated across all study groups on the day of surgery and during the first three postoperative days. Diuresis was significantly higher in Groups 1 and 2, only on the day of the operation, suggesting that dexmedetomidine may promote enhanced urinary output (Figure 4).

While the diuresis values showed an increasing trend in dexmedetomidine-treated groups on postoperative days (POD) 1, 2 and 3 (Figure 4), these differences did not reach statistical significance (*p* = 0.075). GFR remained stable across groups, and no significant differences were found in creatinine and urea levels, indicating no adverse renal effects associated with dexmedetomidine administration (Table A1, Table A2, Table A3 and Table A4).

### 3.5. Postoperative Complications

The incidence of CAM-positive delirium on postoperative day 1 was highest in Group 0 (control) and lowest in Group 2, suggesting a potential protective effect of higher-dose dexmedetomidine (*p* = 0.034). CAM-positive delirium cases were observed in all groups over the following days but were no longer detected from postoperative day 4 onward. The total number of CAM-positive cases was highest in the control group; however, this difference did not reach statistical significance (*p* = 0.155) (Table 3).

Postoperative atrial fibrillation (AF) was detected in all groups, with an equal incidence in Group 0 (control) and Group 1, while the lowest incidence was observed in Group 2. Observed difference was not statistically significant (*p* = 0.412) (Table 3).

Postoperative infections were documented in all groups. Differences among groups did not reach statistical significance (*p* = 0.251) (Table 3). The majority of infections were urinary tract infections, with one case of respiratory tract infection. No surgical site infections were observed. One patient developed late prosthetic valve endocarditis, but this occurred outside the study period. The same patient also contracted COVID-19 during hospitalization. Antibiotic use did not correlate with infection rates, as antibiotics were prescribed at the discretion of the operating surgeon, often without prior microbiological confirmation. Ciprofloxacin was the most commonly prescribed agent, followed by meropenem.

## 4. Discussion

The inflammatory response following CPB is characterized by the release of pro-inflammatory cytokines, particularly IL-6 and TNF-α. The aim of this study was to investigate the effects of dexmedetomidine on inflammatory markers, clinical outcomes, and postoperative recovery in patients undergoing SAVR requiring CPB. Our results highlight the potential immunomodulatory and clinical benefits of dexmedetomidine but also demonstrate that some previously reported effects could not be readily reproduced in our cohort.

The inflammatory response to CPB was confirmed by a significant increase in IL-6 and TNF-α levels in all patient groups postoperatively. Although dexmedetomidine was expected to attenuate this inflammatory surge, our results were mixed. Group 2 (high-dose dexmedetomidine) exhibited the highest postoperative IL-6 levels, though the differences between groups were not statistically significant. However, TNF-α levels were significantly higher in Group 2 at 12 h post-CPB. A notable finding was the correlation between the prolonged CPB duration in Group 2 and the elevated TNF-α levels 12 h post-CPB. This suggests that prolonged extracorporeal circulation may attenuate the expected immunomodulatory effects of dexmedetomidine, despite the administration of a high dose of the studied drug in this group. These findings support the notion that the magnitude of systemic inflammation is multifactorial, influenced not only by pharmacological interventions but also by surgical complexity and perfusion-related factors.

The significantly longer CPB duration in Group 2 warrants discussion, as CPB time varies even within the same surgical procedure. In SAVR, multiple factors influence CPB duration, including surgical complexity and the degree of valve calcification. Extensive calcification necessitates more thorough decalcification, prolonging both operative and CPB times. Despite a standardized surgical protocol, surgeon experience, intraoperative decisions, and technical variability (e.g., speed of aortic cross-clamping, cardioplegia administration, and suture placement) can introduce differences in CPB duration. Additionally, despite randomization, the small sample size may have led to an uneven distribution of surgical complexity across groups. While CPB time differences were observed, they are likely multifactorial, influenced by patient-specific factors, intraoperative variability, and perfusion-related decisions rather than dexmedetomidine administration.

In our study, the neutrophil-to-lymphocyte ratio (NLR) was analyzed as part of the baseline characteristics, as peripheral blood differential counts were only available upon hospital admission. NLR reflects the balance between innate immune activation (neutrophils) and adaptive immune suppression (lymphocytes). No statistically significant differences in NLR were observed between groups at baseline, suggesting that preoperative systemic inflammatory status was comparable across all groups. While IL-6 and TNF-α were the primary cytokines analyzed in this study, NLR could serve as a valuable inflammatory marker in the context of CPB, reflecting perioperative stress, immune suppression, and postoperative outcomes. Given its cost-effectiveness and widespread clinical availability, NLR may be a useful complementary biomarker for assessing systemic inflammation in future trials.

The incidence of postoperative delirium (CAM-positive cases) was highest in the control group and lowest in Group 2 on POD1. Although the overall incidence of delirium did not reach statistical significance, a trend was observed, aligning with previous reports on the neuroprotective role of dexmedetomidine. The reduction in delirium may be attributed to the sedative effect of dexmedetomidine, which preserves sleep architecture and reduces opioid consumption.

In contrast, the incidence of AF was numerically lower in Group 2, but this difference did not reach statistical significance. Considering the established role of systemic inflammation in AF development, the observed trend may reflect the sympatholytic effects of dexmedetomidine. However, the lack of correlation between TNF-α reduction and AF incidence suggests that other inflammatory mediators or hemodynamic factors may be involved. Future research should clarify whether dexmedetomidine reduces AF risk primarily through autonomic modulation rather than anti-inflammatory mechanisms.

Despite the opioid-sparing effect observed in Group 2, the time to extubation was longest in this group. One potential explanation is that higher doses of dexmedetomidine led to dose-dependent bradycardia and hemodynamic depression, necessitating prolonged postoperative monitoring before extubation. This aligns with prior studies indicating that higher dexmedetomidine doses may prolong sedation in cardiac surgery patients. Additionally, most extubation procedures were delayed due to nursing staff shift changes. As a result, the recorded extubation times may not accurately reflect the patients’ actual clinical readiness for extubation. This procedural factor may have masked any potential dexmedetomidine-related effect on early extubation.

Bradycardia incidence was low across all groups, with no significant differences. TCP was required in only one patient in Group 2, while pacing was more frequent in Groups 0 and 1. These findings suggest that dexmedetomidine did not increase the risk of clinically significant bradycardia within the studied dose range. Alternatively, the need for pacing in Groups 0 and 1 may have been influenced by factors unrelated to dexmedetomidine. While our results do not establish a causal relationship between dexmedetomidine and bradycardia, they underscore the need for careful dose titration in hemodynamically vulnerable patients.

Interestingly, the infection rate was higher in the dexmedetomidine-treated groups, although this difference was not statistically significant. This finding contrasts with prior studies suggesting that dexmedetomidine may enhance immune function. A plausible explanation is that the prolonged CPB duration in Group 2 led to greater endothelial dysfunction and immunosuppression, overshadowing any potential protective effect of dexmedetomidine. Antibiotic use did not correlate with infection rates, as antibiotic administration was often based on the operating surgeon’s clinical judgment rather than microbiological confirmation.

Postoperative renal function assessment indicated that dexmedetomidine had no adverse impact on renal function, as GFR and creatinine levels remained stable across all groups. Notably, diuresis was significantly increased in Groups 1 and 2, particularly on POD1, suggesting that dexmedetomidine may enhance fluid excretion and natriuresis. This observation is consistent with prior research suggesting that dexmedetomidine modulates renal perfusion and sympathetic tone, facilitating urinary excretion without impairing renal function. Despite this increased diuresis, other renal markers, including urea and creatinine, did not differ significantly between groups. This supports the assumption that dexmedetomidine does not contribute to renal dysfunction. Concerns regarding dexmedetomidine-induced hemodynamic instability or reduced renal perfusion due to bradycardia were not substantiated, as no evidence of renal impairment was observed. Future investigations involving larger cohorts and additional renal biomarkers (e.g., neutrophil gelatinase-associated lipocalin [NGAL] or cystatin C) may provide a more comprehensive assessment of dexmedetomidine’s potential nephroprotective effects in CPB patients.

This study has several limitations. The relatively small sample size limited statistical power, particularly for detecting differences in AF incidence and infection rates. Additionally, while dexmedetomidine’s anti-inflammatory effects were partially demonstrated, the results may have been confounded by surgical complexity and perfusion times. Future research should involve larger patient populations and expanded inflammatory marker panels, including IL-10, CRP, and nuclear factor kappa B (NF-κB) pathway analysis, to better elucidate dexmedetomidine’s immunomodulatory properties in cardiac surgery patients.

## 5. Conclusions

Dexmedetomidine demonstrated a dose-dependent effect on TNF-α modulation, delirium incidence, and opioid consumption; however, its influence on IL-6 levels and extubation time was less pronounced. These results suggest that while dexmedetomidine remains a valuable adjunct in cardiac anesthesia, its role in modulating systemic inflammation and preventing cardiac arrhythmias warrants further investigation. Careful dose selection is essential, as higher doses may prolong sedation and delay extubation. Future studies should focus on the long-term effects of dexmedetomidine on postoperative recovery and immune function in cardiac surgery patients.

## Figures and Tables

**Figure 1 life-15-00524-f001:**
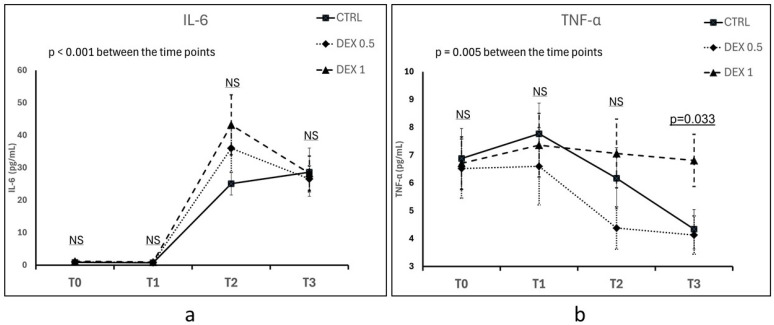
Comparison of IL-6 (**a**) and TNF-α (**b**) concentrations between groups across different time points (T0—hospital admission, T1—prior to CPB initiation, T2—5 h after CPB discontinuation, T3—12 h after CPB discontinuation). Plasma concentrations of IL-6 and TNF-α were measured using enzyme-linked immunosorbent assay (ELISA). Data are presented as the mean ± standard error of the mean (SEM). NS—not significant. Differences between groups and the effects of time were assessed using repeated measures ANOVA.

**Figure 2 life-15-00524-f002:**
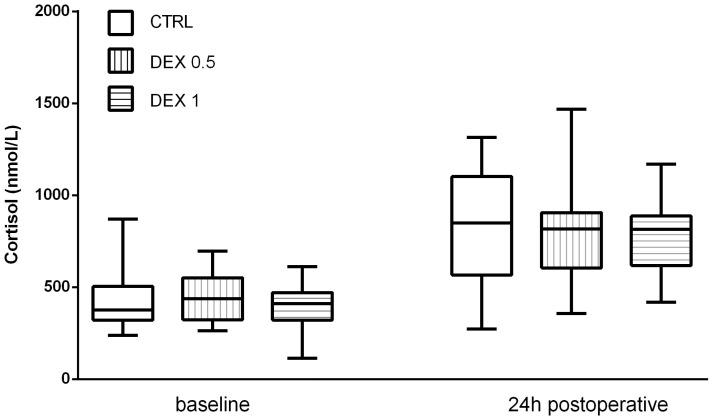
Cortisol levels at baseline and 24 h postoperatively. Data are presented as median, interquartile range with the minimum and maximum values.

**Figure 3 life-15-00524-f003:**
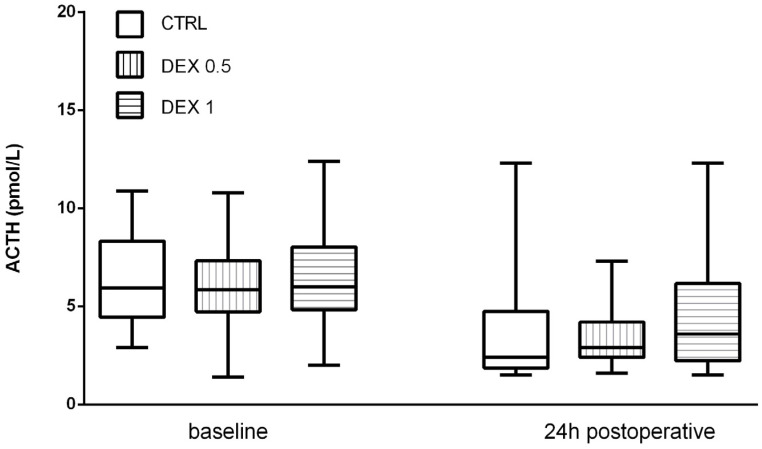
ACTH levels at baseline and 24 h postoperatively. Data are presented as median, interquartile range with the minimum and maximum values. ACTH—adrenocorticotropic hormone.

**Figure 4 life-15-00524-f004:**
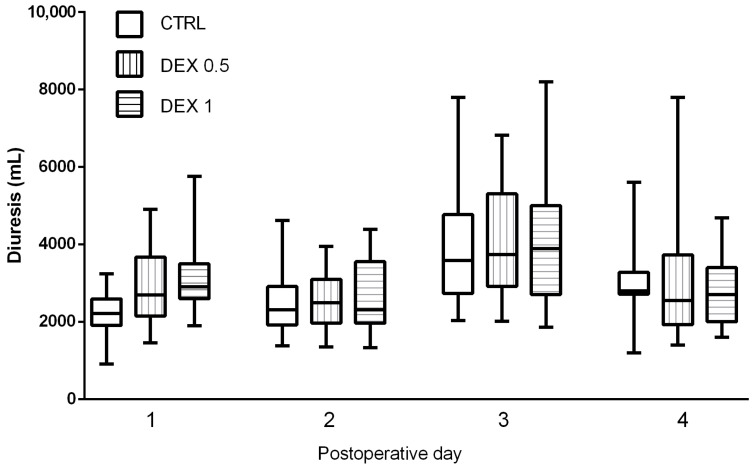
Comparison of diuresis (urine output in mL) between groups during postoperative days (POD) 0–3. Data are presented as median, interquartile range with the minimum and maximum values. Postoperative day 0 (POD 0) refers to the day of surgery, while POD 1–3 represent subsequent days following surgery.

**Table 1 life-15-00524-t001:** Preoperative patient characteristics.

Characteristic	Group 0, *n* = 20	Group 1, *n* = 20	Group 2, *n* = 20	*p*-Value
Age (years)	67.8 ± 8.45	67.15 ± 7.13	68.4 ± 6.44	0.867
Sex				
Male	14 (70%)	16 (80%)	8 (40%)	0.025
Female	6 (30%)	4 (20%)	12 (60%) ^a^	
BMI (kg/m^2^)	28.35 ± 2.02	28.4 ± 2.03	27.93 ± 2.67	0.772
IL-6 (pg/mL)	0.86 ± 0.77	1 ± 0.71	1.2 ± 1.2	0.51
TNF-α (pg/mL)	6.88 ± 4.71	6.51 ± 4.63	6.71 ± 4.12	0.967
GFR (mL/min/1.73 m^2^)	81.32 ± 19.03	82.21 ± 18.87	74.8 ± 16.43	0.376
Creatinine (μmol/L)	79.4 ± 22.25	84.15 ± 35.6	78.1 ± 15.3	0.738
Urea (mmol/L)	7.12 ± 2.42	7.1 ± 2.98	7.54 ± 2.44	0.831
CRP (mg/dL)	2.0 (0.6–4.2)	1.95 (1.0–3.4)	1.5 (1.1–4.75)	0.879 ^b^
PCT (ng/mL)	0.04 ± 0.03	0.05 ± 0.02	0.04 ± 0.02	0.462
NLR	2.269 ± 0.984	3.25 ± 1.66	3.22 ± 1.38	0.286

BMI—body mass index, GFR—glomerular filtration rate, CRP—C-reactive protein, PCT—procalcitonin, NLR—neutrophil-to-lymphocyte ratio. For all variables ANOVA followed by LSD except (^a^) chi squared, (^b^) Kruskal–Wallis followed by Mann–Whitney. Continuous variables are presented as mean ± standard deviation (SD) and categorical data are presented as number of cases (*n*) with corresponding percentages (%).

**Table 2 life-15-00524-t002:** Intergroup differences in opioid consumption, CPB duration, aortic cross-clamp time, and the need for TCP.

Variable	Group 0, *n* = 20	Group 1, *n* = 20	Group 2, *n* = 20	*p*-Value
Sufentanil (intraop) (μg)	93 ± 31.93	87.5 ± 15.17	91.25 ± 18.63	0.744 ^a^
CPB time (min)	64.55 ± 10.86	58.5 ± 11.78	68.65 ± 14.74 ^b^	0.044 ^a^
Cross clamp time (min)	47.85 ± 8.56	44.30 ± 10.08	48.75 ± 12.68	0.378 ^a^
TCP	3 (15%)	3 (15%)	1 (5%)	0.68 ^b^
Morphine to extubation (mg)	17.25 ± 3.8	14.25 ± 6.13	10.42 ± 6.82 ^a,b^	0.002 ^a^
Time to extubation (h)	8.38 ± 3.58	8.35 ± 2.81	9.94 ± 3.91	0.261 ^a^

CPB—cardiopulmonary bypass, TCP—temporary cardiac pacing, (^a^) ANOVA followed by LSD, (^b^) chi squared. Continuous variables are presented as mean ± standard deviation (SD) and categorical data are presented as number of cases (*n*) with corresponding percentages (%).

**Table 3 life-15-00524-t003:** Postoperative complications.

Characteristic	Group 0, *n* = 20	Group 1, *n* = 20	Group 2, *n* = 20	*p*-Value
AF	8 (40%)	8 (40%)	5 (25%)	0.412
Infection	1 (5%)	3 (15%)	4 (20%)	0.251
Antibiotic	3 (15%)	6 (30%)	4 (20%)	0.85
CAM POD1 positive	5 (25%)	3 (15%)	0	0.034
CAM POD2 positive	4 (20%)	1 (5%)	2 (10%)	0.469
CAM POD3 positive	0	1 (5%)	1 (5%)	0.667
CAM positive total	6 (30%)	3 (15%)	2 (10%)	0.155

AF—atrial fibrillation, CAM—Confusion Assessment Method, POD—postoperative day. Statistical analyses were performed using the Chi-squared test and the Linear-by-Linear Association test. Data are presented as the number of cases (*n*) with corresponding percentages (%).

## Data Availability

Data are available from the corresponding authors upon reasonable request.

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
