# Peer review of "Effect of Dexmedetomidine on Cardiopulmonary Bypass Induced Inflammatory Response in Patients Undergoing Aortic Valve Replacement"

_life, 2025, doi:10.3390/life15040524_

Round 1

Reviewer 1 Report

Comments and Suggestions for Authors

I have reviewed the manuscript entitled “Effect of Dexmedetomidine on Cardiopulmonary Bypass Induced Inflammatory Response in Patients Undergoing Aortic Valve Replacement.” The authors are to be congratulated for executing a well-designed prospective randomized controlled trial. This study is an excellent hypothesis-generating work that addresses an important clinical issue in cardiac surgery.

Below are my comments for the authors’ consideration:

Major Comments

  1. Figure 1(b) – Discrepancy in p-Values: The figure legend indicates “p > 0.05 between the groups,” yet the manuscript text repeatedly emphasizes a statistically significant difference at T3 (p = 0.033) for TNF-α levels. The authors should revise Figure 1(b) to clarify this inconsistency—either by updating the p-value annotations or by providing a more detailed explanation in the legend and text.

  2. Interpretation of CPB Time Differences: The results show that Group 1 has a lower CPB time (58.5 ± 11.78 minutes) compared to Group 0 (64.55 ± 10.86 minutes) and Group 2 (68.65 ± 14.74 minutes), which seems counterintuitive. A detailed discussion or additional analysis to explain this finding is warranted.

Minor Comments

  1. Presentation of Figure 2: It is recommended to recreate Figure 2 as boxplots. Boxplots would better illustrate the distribution, medians, and variability of diuresis data compared to the current format.

  2. Sex Distribution Inconsistency: While the text states that there is no statistically significant difference in sex distribution among the groups, Table 1 reports a p-value of 0.025. This discrepancy should be clarified.

  3. Clarity in the Results Section: To enhance readability, it would be helpful to reiterate the differences between Groups 0, 1, and 2 at the beginning of the Results section. This would provide the reader with a clear framework for interpreting subsequent data.

  4. Presentation of Table 3: Converting Table 3 into a figure similar to Figure 1 may improve clarity. A visual representation can often facilitate a quicker and more intuitive comparison of hormonal and inflammatory markers across groups.

  5. Language Regarding “Not Statistically Significant” Findings: The manuscript contains several phrases such as “numerically lower” (e.g., for atrial fibrillation incidence and TNF-α levels) and “tended to be lower” when differences do not reach statistical significance. Given the pre-specified alpha of 0.05, these expressions can be misleading. The authors should use definitive language that aligns with their statistical findings. For instance, if a difference is not statistically significant, it should simply be stated as such without implying a trend that the data might not support.

In summary, while the study offers valuable insights into the immunomodulatory effects of dexmedetomidine in cardiac surgery, addressing the issues outlined above will improve the clarity and impact of the manuscript. 

Reviewer 2 Report

Comments and Suggestions for Authors

Thank you for the opportunity to review this intriguing article investigating the effect of dexmedetomidine on cardiopulmonary bypass (CPB)-induced inflammation following surgical aortic valve replacement (AVR). This is a single-centre, prospective, double-blinded, randomized controlled trial involving 60 patients randomized into three groups: a control group receiving saline, and two treatment groups receiving dexmedetomidine at either 0.5 μg/kg/h or 1 μg/kg/h.

The primary outcomes assessed were inflammatory markers (IL-6 and TNF-α), opioid consumption, postoperative delirium, renal function, and incidence of atrial fibrillation. Key findings include significantly reduced morphine consumption, lower incidence of postoperative delirium and atrial fibrillation, increased TNF-α levels at 12 hours post-CPB, and enhanced postoperative diuresis in the high-dose dexmedetomidine group.

The study is well-designed and addresses an important clinical question with considerable practical relevance. However, there are several limitations, such as a relatively small sample size and the absence of a power calculation, limiting the robustness of conclusions, especially regarding clinical endpoints like delirium and atrial fibrillation. Furthermore, variability in CPB duration—particularly prolonged duration in the high-dose dexmedetomidine group—could confound the inflammatory response results. Future studies should address these methodological issues, potentially through larger, multi-centre trials.

Current evidence generally supports dexmedetomidine’s role in reducing pro-inflammatory cytokines, oxidative stress, and sympathetic activation associated with CPB. However, this study reported an unexpected increase in TNF-α levels at 12 hours post-CPB with high-dose dexmedetomidine. It would be valuable for the authors to discuss potential mechanisms or hypotheses explaining this finding in contrast to existing literature.

Overall, despite its limitations, this study provides meaningful insights and positively contributes to the existing literature.

Reviewer 3 Report

Comments and Suggestions for Authors

The article entitled "Effect of Dexmedetomidine on Cardiopulmonary Bypass Induced Inflammatory Response in Patients Undergoing Aortic Valve Replacement" is interesting taking into consideration the importance of dexmedetomidine use in patients undergoing a cardiac surgery. I would have some suggestions for the authors:

1. I would also include the neutrophile/lymphocite ratio since it is a very easy to use parameter. 

2. I don't understand exactly what the tables from the appendix represent. Are those parameters in evolution/postoperatively? If so, they should also be included in the results section and have a short explanation. It is important to see those parameters in evolution since this is the main subject of the manuscript. 

3. What about pulmonary/neurological status before the surgery? And other comorbidities (COPD/stroke/anemia/ etc?). All of them have an important impact on how a cardiac surgery will proceed.

4. What about other treatments that the patients received postoperative? Could they influence the results? I would also include all important therapies that the patients received (not just supportive treatment).

5. How long were the patients observed?

Round 2

Reviewer 3 Report

Comments and Suggestions for Authors

The authors responded to all of my questions. The article is suitable for publication.